# Identifying and Avoiding Risk of Bias in Caries Diagnostic Studies

**DOI:** 10.3390/jcm10153223

**Published:** 2021-07-22

**Authors:** Jan Kühnisch, Mila Janjic Rankovic, Svetlana Kapor, Ina Schüler, Felix Krause, Stavroula Michou, Kim Ekstrand, Florin Eggmann, Klaus W. Neuhaus, Adrian Lussi, Marie-Charlotte Huysmans

**Affiliations:** 1Department of Conservative Dentistry and Periodontology, University Hospital, Ludwig-Maximilians University Munich, 80336 Munich, Germany; svetlana.kapor@yahoo.com; 2Department of Orthodontics and Dentofacial Orthopaedics, University Hospital, Ludwig-Maximilians University Munich, 80336 Munich, Germany; mila_janjic@yahoo.com; 3Department of Orthodontics, Section of Preventive and Paediatric Dentistry, University Hospital, 07743 Jena, Germany; Ina.Schueler@med.uni-jena.de; 4Clinic for Operative Dentistry, Periodontology and Preventive Dentistry, University Hospital RWTH Aachen, 52074 Aachen, Germany; fkrause@ukaachen.de; 5Department of Odontology, University Copenhagen, 2200 Copenhagen, Denmark; stmi@sund.ku.dk (S.M.); kek@sund.ku.dk (K.E.); 6Department of Periodontology, Endodontology and Cariology, University Center for Dental Medicine UZB, University of Basel, 4058 Basel, Switzerland; florin.eggmann@unibas.ch (F.E.); klaus.neuhaus@unibas.ch (K.W.N.); 7School of Dental Medicine, University of Bern, 3012 Bern, Switzerland; adrian.lussi@zmk.unibe.ch; 8Department of Operative Dentistry and Periodontology, University Medical Centre, 79098 Freiburg, Germany; 9Department of Dentistry, Radboud University Medical Centre, 6525 Nijmegen, The Netherlands; Marie-Charlotte.Huysmans@radboudumc.nl

**Keywords:** caries detection, caries diagnostics, caries assessment, caries monitoring, methodology, laboratory studies, clinical studies, reference standard

## Abstract

Caries diagnostic studies differ with respect to their design, included patients/tooth samples, use of diagnostic and reference methods, calibration, blinding and data reporting. Such heterogeneity makes comparisons between studies difficult and could represent a substantial risk of bias (RoB) when it is not identified. Therefore, the present report aims to describe the development and background of a RoB assessment tool for caries diagnostic studies. The expert group developed and agreed to use a RoB assessment tool during three workshops. Here, existing instruments (e.g., QUADAS 2 and the Joanna Briggs Institute Reviewers’ Manual) influenced the hierarchy and phrasing of the signalling questions that were adapted to the specific dental purpose. The tailored RoB assessment tool that was created consists of 16 signalling questions that are organized in four domains. This tool considers the selection/spectrum bias (1), the bias of the index (2) and reference tests (3), and the bias of the study flow and data analysis (4) and can be downloaded from the journal website. This paper explores possible sources of heterogeneity and bias in caries diagnostic studies and summarizes the relevant methodological aspects.

## 1. Introduction

Numerous in vitro and in vivo caries-lesion detection, diagnostic, assessment and/or monitoring studies have been designed, conducted and published to describe the diagnostic accuracy of test methods in terms of validity and reliability within the last several decades. Most recently, several systematic reviews and meta-analyses merged the available data and drew conclusions for clinical practice, e.g., Refs. [1,2,3,4]. In these reports, the author teams consistently mentioned substantial heterogeneity between the studies, which can be predominantly linked to the qualitative strength or shortcomings of a study. The quality of a diagnostic study is heavily influenced by its design, the recruited patient and/or tooth sample, the methods and standards that are employed, the blinding between the diagnostic and reference tests, the statistical analysis and the data reporting. As heterogeneity is directly correlated to the risk of bias (RoB), the assessment of RoB is considered an integral part of systematic reviews. The method commonly used to examine the quality of diagnostic studies is the QUADAS 2 judgement tool (Quality Assessment of Diagnostic Accuracy Studies), which was developed from the initially published QUADAS tool [5,6]. The QUADAS 2 checklist defines study/report areas and provides predefined criteria for assessing the RoB for each area. While planning and conducting systematic literature reviews and meta-analyses of caries diagnostic trials, our study group aimed to include the QUADAS 2 tool for quality assessment. When initially testing this tool, it became obvious that QUADAS 2 was developed primarily for in vivo diagnostic studies in medicine and, therefore, did not fit the specific methodology of caries diagnostic studies, which often focus on lesion detection and/or assessment and are mostly performed under laboratory conditions. Therefore, the aim of this development effort was to create a tailored RoB assessment tool for caries diagnostic studies.

## 2. Materials and Methods

Members of the expert group were selected based on their scientific expertise in conducting caries diagnostic studies, and the group was initiated by two group members (J.K., K.N.). All experts were invited by email in mid-2017. I.S. had the leading role in organizing the financial support for the meetings, which resulted in funding by the German Research Foundation (Deutsche Forschungsgemeinschaft, SCHU-3217/1-1).

To develop recommendations for RoB assessment in caries diagnostic studies, a set of criteria was composed, discussed and agreed upon during three workshops conducted between October 2017 and March 2019. Initially, the group drafted a preliminary criteria sequence that formed the basis for the discussions during the workshops as well as the writing of the final document. Here, existing RoB assessment tools, e.g., the QUADAS 2 tool [5,6] and the Joanna Briggs Institute Reviewers’ Manual [7], influenced the hierarchy and phrasing of the signalling questions that were adapted to the specific dental purpose. The development of the RoB tool was advanced further by the parallel performance of systematic reviews of the literature on diagnostic studies of proximal- and occlusal-surface caries. Here, the author group pre-tested, discussed, improved and circulated the preliminary RoB criteria for several rounds until a major consensus was reached. The compiled draft was made available to the expert group, and the final version was consented to by the expert group during their workshops. In addition to the development of the tailored document for RoB assessment (Appendix A), this manuscript was drafted to provide background information about potential sources of bias that should ideally be ruled out from the beginning when designing future studies. The RoB assessment checklist can be downloaded as a supplementary table from the journal website.

## 3. Results

The tailored RoB assessment tool for caries diagnostic studies (see Appendix A) is structured into four domains with a total of 16 items. A detailed description is given in the following paragraphs.

### 3.1. Domain 1: Selection and Spectrum Bias

In each diagnostic study, the participants and/or teeth should be characteristic—or optimally representative—of the patients for whom the results of the study will be needed. An estimated and adequately powered sample size chosen at random might be considered key features for limiting the RoB. To our knowledge, representative, population-based samples have rarely been used; the same is true for true random selection of patients or teeth. The selection of patients and teeth usually varies with the research question and chosen study type (Table 1).

*Patient selection in clinical studies (Item 1).* In cases where a study of clinical diagnostic accuracy is conducted, the patient sample should be selected based on predefined inclusion and exclusion criteria, aiming to enroll a homogenous sample of individuals, e.g., children, adolescents, adults, or senior adults, in order to estimate the proportion of different caries stages by at least one diagnostic test method. In such epidemiological studies, rigorous validation is not applicable, and bitewing radiography is sometimes considered an additional diagnostic method, e.g., Refs. [8,9]. Currently, strict ethical rules limit study-related X-ray exposure [10], and therefore, this modality is now rarely used in epidemiological or clinical studies. Another substantial drawback in clinical diagnostic accuracy studies is that a rigorous validation is limited to cases that require restorative dental care only, which is, furthermore, indicated less frequently in populations with a low caries risk. Thus, a representative study of clinical diagnostic accuracy with rigorous validation is most likely hard to conduct. Another key feature of good scientific practice—randomization of patients and/or teeth—is also difficult to implement in studies of clinical diagnostic accuracy due to the requirement for independent clinical screening before investigation. To our knowledge, no study of clinical diagnostic accuracy has enrolled a random sample so far.

*Tooth selection (Item 2).* The selection of specific tooth types should be guided by the typical clinical requirements. Proximal caries detection is mostly required for posterior teeth, and the same applies for occlusal caries detection with an emphasis on molar teeth (premolars may be included but should be analysed separately). For smooth surface caries, all types of teeth may be suitable for inclusion. The inclusion of homogenous study material seems to also be of relevance in laboratory studies, and the condition of extracted teeth should be assessed during screening. Existing restorations, caries in locations other than the sites/surfaces under study, developmental disorders, extensive tooth wear, internal discolorations, incomplete root formation or extraction damage may indicate a potential source of bias, and such teeth should be excluded. Although the exclusion of these complicating features may reduce clinical relevance and the diagnostic systems may be clinically required to be robust in such circumstances, this approach will improve the standardization and reduce heterogeneity between studies. As storage conditions have been shown to affect certain diagnostic techniques [11], the storage time and conditions need to be reported. Studies that fulfil the abovementioned characteristics can probably be judged to have a low RoB.

Most studies of caries diagnostic accuracy are performed in vitro using extracted human teeth. In such studies, teeth are either selected from an existing collection or, less commonly, collected expressly for the study. As far as possible, relevant details about the clinical background, such as the population type or reasons for extraction, should be considered and reported. Premolars and third molars extracted for orthodontic reasons in adolescents are most commonly available but might have different diagnostic test accuracies than first and second molars and premolars extracted in older adults for periodontal reasons. Homogeneous tooth selection seems to lower heterogeneity. Selecting a random sample of target teeth has not been considered so far and might also be a sophisticated procedure, especially under clinical circumstances, as this method would require an independent and, optimally, blinded pre-examination of all patients and teeth. Nevertheless, the procedure can be integrated much more easily in an in vitro study design if enough tooth material is available.

*Spectrum of caries lesions (Item 3).* With respect to the chosen research question, different tooth sampling strategies can be considered for clinical and laboratory studies: (1) balanced sampling of all caries lesion types, (2) sampling related to the probable population-based caries distribution or (3) selective sampling of distinct caries lesions. Moreover, tests of the overall diagnostic accuracy of a distinct method that ideally covers the whole caries spectrum may be proposed. This requires a pre-evaluation (e.g., visually and/or radiographically) and a separate evaluation step aiming to safeguard a well-balanced sample with nearly equal proportions of healthy, non-cavitated and cavitated caries lesions. This approach should enable the researcher(s) to identify strengths and limitations of the test method in relation to all relevant diagnostic categories. Furthermore, the exclusion or underrepresentation of stages of the caries process may lead to distortion of the diagnostic accuracy results. However, it has been quite common in studies of caries diagnostic accuracy to eliminate progression stages at the most severe end of the spectrum, such as gross cavitation. Although eliminating this stage may be expected to lead to an underestimation of diagnostic accuracy as “easy” cases are avoided, excluding them may be justified for reasons of clinical relevance in most populations with a low caries risk, where such severe lesions may be rare. Following this approach, the caries spectrum sampled could be matched in relation to the probable target population in which the diagnostic system is expected to be used [11,12,13,14]. As such, different sampling strategies are expected to result in different outcomes, and reporting on the used caries spectrum is mandatory. In studies where diagnostic performance is expressed as a correlation of the method outcome with the caries lesion depth or stage, an even distribution of the sample over the entire caries spectrum seems to be required. Eliminating potential spectrum bias in studies of clinical diagnostic accuracy is especially challenging. Here, at its best, only one caries lesion or examination site from each patient should be validated to avoid an overrepresentation of a single subject. Alternatively, corrections for clustering of data might be considered in the statistical analysis. However, it must be noted that, so far, it is unknown what effects the previously mentioned strategies aiming to reduce spectrum bias might have on diagnostic accuracy. Therefore, future studies should consider such methodological aspects.

*Sample size in validation studies (Item 4).* Sample size calculations have been almost universally missing from caries diagnostic reports. Perhaps this is an oversight, but calculations show that for a traditionally composed in vitro sample of teeth (complete caries spectrum, sound sites and sites with caries lesions in enamel and into dentin present) with about half of the sample not having caries extending into the dentin, the required number of teeth needed to calculate both sensitivity (SE) and specificity (SP) to a confidence level of +/− 10% easily exceeds 100 teeth. Most caries diagnostic studies struggle to reach this sample size, and the confidence level may not be clinically acceptable (for an estimated SE, it may be somewhere between 60% and 80%). The sample size in clinical studies may be even more challenging, as the disease prevalence may not be manipulated as it is with in vitro samples, and the prevalence is typically low. Sample size calculations, or at the least post hoc power calculations, should be reported.

In the case of diagnostic accuracy studies, which aim to calculate SE and SP, a statistical procedure for sample size calculation was suggested by Burderer [15]. The precision of SE and SP is influenced by the sample size and the prevalence of the disease. Therefore, the study sample size should be sufficient to ensure that the study will lead to the precise calculation of SE and SP. In cases where a small number of research subjects is used in a study or the prevalence is different from the population to whom the diagnostic test will be applied, a low-precision SE and SP might be calculated, and thus the result might not be clinically applicable or might provide misleading information. On the other hand, a larger number than necessary might lead to a waste of resources. The following formulas can be used for the calculation at a specific required absolute precision level [15,16]:-Sample size (*n*) based on SE: $n=\frac{Z_{\alpha /2}^{2}\times SE\times (1-SE)}{L^{2}\times Prevalence}$Sample size (*n*) based on SP: $n=\frac{Z_{\alpha /2}^{2}\times SP\times (1-SP)}{L^{2}\times (1-Prevalence)}$

*n* is the required sample size; SE and SP are the estimated SE and SP values for the investigated diagnostic method based on previous literature or pilot studies; α defines the confidence level (CI = 1 − α) used in the study and is usually set at 0.05 or 0.01; *L* is the desired absolute precision on both sides of the SE and SP outcomes (the maximum clinically acceptable width of the CI) and in this type of validation study for diagnostic methods is suggested to be ≤0.1; Z_α/2_ is a constant that corresponds to different CIs and can be found in common statistical textbooks or online formulas (Table 2). Monograms or tables based on this formula are also available in the literature and can be used for sample size estimation [15,16,17,18].

*Sample size in reliability studies (Item 4).* The required sample size in reliability studies is even less clear. The majority of reliability data that have been published so far use a pre-test sample or sub-sample taken from the whole study material. This may limit the generalizability, and future studies should include an appropriate sample size [19].

### 3.2. Domains 2 and 3: Index and Reference Test

*Index test methodology (Item 5).* The test methodology has to be explicitly reported to evaluate its proper conduct. For most diagnostic test methods, several scoring criteria and/or thresholds are being used, which may limit the comparability between studies. Diagnostic criteria of the index test and their corresponding reference standard (Table 3) have to be carefully justified according to the study aim and must be described in detail. The key component of diagnostic accuracy studies is the rigorous validation of the index tests with an appropriate reference test (gold standard). Here, it is essential that both the index tests and the reference tests use plausible thresholds; otherwise, the results might be biased.

*Reference test methodology (Item 6, 8–10).* The choice of the reference method and its threshold(s) may vary between studies and, therefore, may potentially induce heterogeneity. The situation is further complicated by the fact that, with changing attitudes towards the indications for restorative treatment, the lesion characteristics considered relevant for treatment decisions are changing, and thus, the threshold criteria for validation are changing. This trend may lead to considerable heterogeneity in the accuracy results.

Rigorous validation of caries extension under in vitro conditions can be performed in all included teeth using different nondestructive techniques, e.g., X-ray microtomography, and/or destructive histological techniques, e.g., hemisectioning, serial sectioning, or grinding. In addition, the caries process needs to be imaged, which can be performed with different techniques, e.g., light microscopy with or without staining or microradiography. In the final step, caries (depth) detection should be described on the basis of well-defined criteria in relation to anatomical tissues/structures, e.g., enamel, enamel–dentin-junction (EDJ), dentin, pulp or treatment-related thresholds. This assessment of histological images and the metric assessment can be performed using semiquantitative categories, e.g., Refs. [20,21] and/or quantitatively, e.g., Refs. [22,23,24]. In principle, it can be assumed that the more histological slices or images that are available, the more precise the determination of the site with the most extended caries lesion would be. Problematically, there is little information published so far regarding which technique(s) could be preferred as a standard procedure, and little is known about the comparability between the mentioned reference standards, e.g., Refs. [22,23,24]. To avoid incorporation bias, the information from any index test method must be excluded as much as possible from the reference test.

While the reference test can typically be performed under laboratory conditions without any restrictions, the situation is more complex in clinical studies. Here, the common problem is the inability to prove the caries characteristics on healthy surfaces or caries lesions that can be treated preventively or with minimally invasive procedures because, any thorough reference standard requires an invasive tissue analysis that cannot be justified due to ethical reasons. Therefore, conducting a diagnostic study under in vivo conditions is typically subject to verification bias due to the simple facts that, first, destructive validation of sound surfaces and non-cavitated caries lesions is contraindicated, and, second, validation by biopsy leads to an imprecise measure in comparison to results from rigorous in vitro validation. Therefore, alternative study designs might be used to overcome both limitations (Table 1). When considering first the difficulty of predicting the exact caries extension in vivo, crude validation by biopsy can be supplemented by an additional radiograph from a replica of the validated tooth to predict caries extension without any health risks [25]. The major limiting aspect—the impossibility of validating all teeth or caries lesions—is hard to overcome. One option could be quasi-validation by diagnostic reassessment a period of time later [26]. Another opportunity might be the inclusion of a pseudo-validation method, e.g., the usage of radiography. Both issues are describing the researcher’s dilemma in clinical diagnostic studies that could not be resolved so far as there is no noninvasive, intraoral reference standard available. Nevertheless, clinical studies will be needed to assess the diagnostic accuracy and practicability; furthermore, some diagnostic test methods are impossible to perform in an unbiased manner under in vitro conditions, underlining the importance of in vivo diagnostic studies. Therefore, strategies need to be developed to lower the heterogeneity between study designs.

Blinding is done between procedures of sample selection, index and reference tests (Items 6, 9 and 10). To reduce the RoB by familiarity with the study sample, all evaluations should be performed blindly and separately from each other. Therefore, appropriate blinding measures should be considered: (1) Most optimally, each relevant step, e.g., pre-evaluation, diagnostic test(s) and reference standard, will be performed by a separate researcher; (2) if overlap is unavoidable, each step has to be performed with a sufficient time lapse (at least one or two weeks) between examinations to reduce memorization by the investigator; and (3) under in vitro conditions, a randomized/shuffled allocation of the specimens or diagnostic images as well as inclusion of dummy teeth may help to reduce memorization by investigators. Performing sufficient blinding in clinical studies is basically possible but complicated, as multiple dentists need to be included when performing clinical evaluations independently from each other.

*Training and calibration (Items 7 and 10).* It is necessary that each member of the study group be qualified and passed through training and (where appropriate) calibration under the supervision of at least one experienced researcher. Such a course should include an independent sample of teeth or individuals who share the same constitution as the test sample and is not part of the main investigation. The extent of the theoretical and practical training depends mainly on the experience level of the trainees. To prove the success and low bias of the training, the intra- and inter-examiner reliability must be determined as an indicator that the examiners were trained sufficiently. Here, the same statistical methods that are recommended for exploring data in reliability studies (see below) can be used. In cases where learning effects should be investigated, extensive training and calibration is typically modified.

### 3.3. Domain 4: Study Flow & Data Analysis

*Verification bias (Items 11 to 14).* All teeth must undergo the same index test and consistently receive the same reference test. Both procedures should be performed separately from each other. In cases where patients withdraw from study participation or are lost to follow-up or in cases where teeth/specimens are unavailable for diagnosis or validation or are lost during the laboratory workflow, this needs to be reported. Fulfilment of these aspects probably indicates a low RoB.

*Validity bias (Item 15).* Until now, 2 × 2 contingency tables (Figure 1) have been widely used to assess the diagnostic accuracy of diagnostic test methods in relation to an appropriate reference standard [27]. Here, it seems to be important that the corresponding threshold values be plausibly chosen for the index and reference test and, finally, dichotomize the data accurately. In addition, some thresholds, e.g., the intervention level for operative dental care, remain a matter of discussion. It is suggested that future studies should provide their diagnostic findings in as detailed a manner as possible. Here, other formats, e.g., a 6 × 6 contingency tabulation (Figure 2), might be considered as possible option to describe the case distribution comprehensively in relation to more than one threshold. On the basis of such a descriptive table, the diagnostic accuracy including SE, SP, negative and positive predictive values or any meta-analysis can be calculated for several thresholds. In addition to contingency tables, the area under the ROC curve (AUC) is frequently used and will remain a valuable statistical tool in diagnostic studies [27].

*Reproducibility bias (Item 16).* To determine the agreement between different researchers, devices or clinical workflows, the data sets generated by at least one examiner will typically be compared to repeated measurements by the same operator (intra-examiner reliability) or other operators (inter-examiner reliability). It might also be possible to include a consensus decision that was made by a study/expert group and to compute the individual decisions against this reference. An opportunity to assess the intra- and inter-examiner reproducibility seems to be the usage of a tooth sample before validation is performed. While blinding has to be safeguarded between different examinations, operators and/or diagnostic courses, the inclusion of calibration training seems to be dependent on the study aim. This is definitely needed in the case of diagnostic validation studies or epidemiological trials when it is essential to provide proof that each examiner has high-level knowledge. Conversely, in studies investigating the possible learning progress of examiners, the inclusion and extent of calibration training may vary. In this context, it is also known that reliability might be affected by the educational level and experience of the observer(s) as well as the quality and discriminative power of the detection method(s). Therefore, these aspects need to be reported to enable a correct interpretation of the results.

Different statistical measures have been recommended and used to explore reliability data; examples include (weighted) Kappa coefficients for categorial data, e.g., Refs. [27,28,29,30,31], correlation coefficients for continuous data, e.g., Ref. [32], Bland–Altman plots [33,34,35], mixed-effect models [36] and logistic regression analysis.

## 4. Discussion and Knowledge Gaps

During the development of the RoB assessment tool (Appendix A), several issues were discussed in the study group that could not be fully resolved. In the following, some examples are summarized. While an appropriate sample size can be calculated on the basis of the described statistical procedure, there are no strict recommendations on the proper sample constitution for validation and reproducibility studies. Another unresolved issue is the question about sufficient blinding procedures. Here, some proposals were made in this summary, but it must be acknowledged that several procedures have not been used so far; therefore, little is known regarding the practicability and the degree of blinding that should be reached. Little is also known about the comparability of existing reference standards for caries validation, and it must be mentioned again that no widely accepted standard procedure is available and that different validation techniques do exist. Another relevant, unresolved problem is the validation of caries activity [39]. The latter aspect also indicates future research needs.

## 5. Conclusions

The present paper has explored possible sources of heterogeneity and bias in caries diagnostic studies and has summarized methodological aspects that could be linked to a high RoB. The tailored assessment tool (Appendix A) provides assistance for a systematic evaluation of RoB, which should be considered by research and author teams, reviewers and editors.

## Figures and Tables

**Figure 1 jcm-10-03223-f001:**
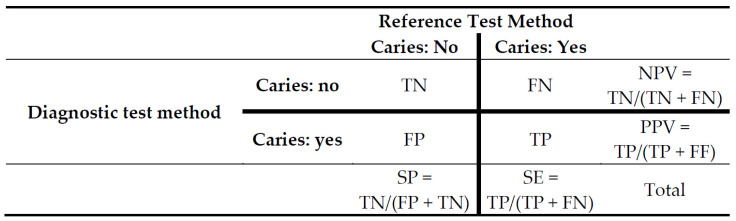
2 × 2 contingency table for determining the validity of a distinct diagnostic test method in relation to a rigorous, mostly histology-based reference standard. The bold printed lines indicate the threshold value. True negative (TN)—The index test is negative, and the reference test is negative. True positive (TP)—The index test is positive, and the reference test is positive. False negative (FN)—The index test is negative, and the reference test is positive. False positive (FP)—The index test is positive, and the reference test is negative.

**Figure 2 jcm-10-03223-f002:**
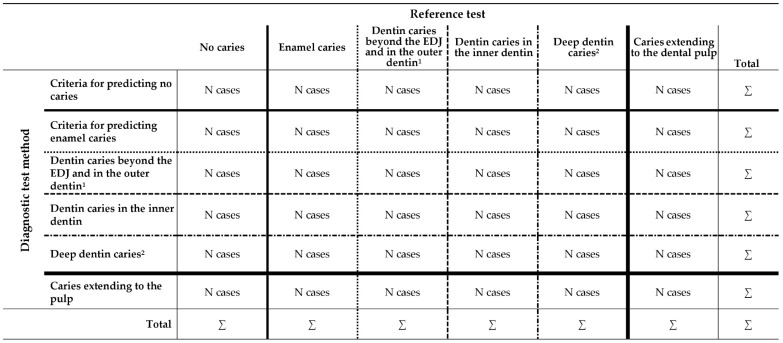
6 × 6 contingency tabulation for potential use in future caries detection, diagnostic, assessment and/or monitoring studies that enables a complete reporting of all caries stages. It is important to recognize that different thresholds need be used for predicting different caries stages, e.g., caries detection level (bold line), dentin caries detection level (dotted line), outer dentin caries level (dashed line), deep dentin caries detection level (dash-dotted line) or caries involving the pulp (jagged line). ^1^ There is no exact definition for the metric extent of the “outer dentin”, and different proposals have been made, e.g., 33% [21] or 2 mm [37]. Other thresholds might also be possible. ^2^ A deep dentin caries lesion reaches the inner quarter of the dentin [38].

**Table 1 jcm-10-03223-t001:** Available types of studies for the investigation of diagnostic test methods for primary caries detection, diagnosis, assessment and/or monitoring.

Type of Study	Study Design	Study Material	Limitations
In vitro diagnostic study with in vitro histological validation	Favoured study design to determine the accuracy of a diagnostic test method/procedure in comparison to a rigorous in vitro reference standard when the in vitro diagnostic test procedure leads to similar diagnostic findings in comparison to the clinical usage; otherwise an in vivo study design has to be chosen.	Inclusion of appropriate human tooth material—non-damaged, non-restored, extracted teeth with different stages of caries lesions on the surface of interest.	Simplified in vitro test procedures in comparison to the clinical situation. Potentially biased spectrum of target teeth and caries lesions due to the unavailability of appropriate human tooth material.
In vivo diagnostic study with in vitro histological validation	After performing diagnostic test(s) of in vivo and tooth extraction, caries lesions will be rigorously examined under in vitro conditions.	Inclusion of only those teeth that will need tooth extraction, e.g., 3rd molars.	Potentially biased spectrum of target teeth and caries lesions, ambitious patient recruitment.
In vivo diagnostic study with in vivo validation	Study design that might be taken into consideration when an in vitro validation is not feasible. Only those proportion of teeth that need operative dental care will be validated, with no validation of healthy surfaces and non-cavitated caries lesions (incomplete in vivo reference standard).	Inclusion of only those patients with teeth that need operative dental care.	Typically, no validation of healthy surfaces and non-cavitated caries lesions.
In vivo diagnostic study with in vivo validation and quasi-validation	Study design that might be taken into consideration when an in vitro validation is not feasible. Validation of only those cases that need well-justified, operative dental care and/or quasi-validation by longitudinal monitoring of healthy surfaces and non-cavitated caries lesions using convincing diagnostic test method(s).	Inclusion of all target patients and teeth.	Less rigorous and mixed in vivo reference standard, longitudinal and ambitious study design.
In vivo diagnostic study without validation by biopsy (with or without usage of a pseudo-reference standard)	Study design to examine/test/analyse the outcome of at least one diagnostic test method. It might also be possible to use a diagnostic test method, e.g., radiography, as replacement for a rigorous in vivo or in vitro validation (pseudo-reference standard).		No validation; the pseudo-reference standards are less powerful than rigorous reference standards.
Caries monitoring studies	Longitudinal study design to investigate changes in caries characteristics or activity with at least one diagnostic test method. No validation is performed.		Longitudinal and ambitious study design.
In vivo or in vitro diagnostic study on reliability	Reliability can be investigated under in vivo or in vitro conditions. The diagnostic test protocol needs to be repeated at minimum once, aiming to determine intra- and inter-examiner reproducibility.		-

**Table 2 jcm-10-03223-t002:** Typically, used Z_α/2_ scores for diagnostic studies.

α-Error (2-Sided)	5%	1%	0.1%
Z_α/2_	1.96	2.5758	3.2905
Power	80%	90%	95%
Z1–β	0.8416	1.0364	1.6449

**Table 3 jcm-10-03223-t003:** Overview of possible caries diagnostic methods and reference standards in relation to dental diagnoses.

Diagnosis	Caries Diagnostic Method(s)	Reference Standard
- Non-cavitated surface/caries - Cavitation in enamel - Cavitation in dentin	Visual and/or visual-tactile	In vitro: Visual inspection/direct microscopy
		In vivo: Visual inspection/direct microscopy (after tooth separation)
- No caries - Caries extension in enamel (caries detection level) - Caries extension beyond the EDJ and in the dentin (dentin caries detection level) - Caries extension beyond the outer dentin, e.g., ^1^/_5_ or ^1^/_3_ dentin involvement (probable threshold for operative intervention) - Caries extension beyond the inner quarter of dentin (deep dentin caries detection level) - Caries extension reaching the dental pulp	Visual, radiography, photo-optical methods, others and its combinations	In vitro: Rigorous histological validationIn vivo: Validation by biopsy, pre- and/or post-operative radiography, etc.
- Caries inactivity - Caries activity	Multifactorial, clinical diagnosis which includes different lesion characteristics, biofilm presences, “age” of the lesion, etc.	In vitro: No standard protocol available so far
		In vivo: No standard protocol available so far

## Data Availability

Not applicable for this type study.

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
