# Peer review of "Identifying and Avoiding Risk of Bias in Caries Diagnostic Studies"

_jcm, 2021, doi:10.3390/jcm10153223_

Round 1

Reviewer 1 Report

The article sent for review entitled "Identifying and avoiding the risk of bias in caries diagnostic studies" is properly prepared for publication in every aspect.

I would like to emphasize that the research methodology was prepared to step by step through a discussion conducted during the Authors of the manuscript meetings. Therefore the methodology that was the basis for the preparation of the current work is perfect. The current manuscript should be not only a guide but a base for caries diagnostic studies both in vitro and in vivo.

It only remains for me to congratulate the Authors for undertaking the preparation of the article which is so necessary for future researchers.

Author Response

Thank you! ?

Reviewer 2 Report

The manuscript is well-written and organised. It also presents very important and up-to-date issue. However, I have some minor remarks:

Lines 64-65 - Please add information who and when (during meeting/conference?) selected members of the expert group. What is the name of the expert group?

Table 1 – I suggest dividing column Study material & Limitations into 2 separate columns Study material and Limitations. You can also merge cells with the same characteristics – it might add clarity.

It is not clear, what do you mean by terms ‘Domain’ and ‘Item’. Please explain before first usage.

Congratulations on this paper.

Author Response

The manuscript is well-written and organised. It also presents very important and up-to-date issue. However, I have some minor remarks:

Lines 64-65 - Please add information who and when (during meeting/conference?) selected members of the expert group. What is the name of the expert group?

Response:          Thank you for this point. We added the information that all experts were invited by email in the middle of 2017.

Revised text:      L68.

Table 1 – I suggest dividing column Study material & Limitations into 2 separate columns Study material and Limitations. You can also merge cells with the same characteristics – it might add clarity.

Response:          Thanks for your remark. We followed your suggestion and re-structured table 1.

Revised text:      Table 1.

It is not clear, what do you mean by terms ‘Domain’ and ‘Item’. Please explain before first usage.

Response:          We added a sentence at the beginning of “Results”.

Revised text:      L89-91.

Congratulations on this paper.

Response:          Thank you! ?

Reviewer 3 Report

Dear Authors, 

I really believe that the article is well written and designed. However, I also think that there is a need to simplify what was written in the results section. I was in difficulty when I read that section probably because I don't have suffient knowledge in statistics. By the way, my only suggestion is to make easier and more flowing that part of the manuscript to better capture the attention of readers

Best regard. 

Author Response

I really believe that the article is well written and designed. However, I also think that there is a need to simplify what was written in the results section. I was in difficulty when I read that section probably because I don't have suffient knowledge in statistics. By the way, my only suggestion is to make easier and more flowing that part of the manuscript to better capture the attention of readers.

Response:          Thank you for reading and reviewing our manuscript. - We carefully re-read the whole manuscript and are were unable to address this point. We deeply believe that the chosen structure guides the reader through all important aspects in conducting caries detection/ diagnostic studies. Further, we are convinced that the paper has explored methodological aspects as well as potential sources of heterogeneity and bias in detail for the first time.